# CoQuEST: Entity-Focused Code-Mixed Question Generation for Entertainment Videos

## Abstract

Earlier research on video-based question generation has primarily focused on generating questions about general objects and attributes, often neglecting the complexities of bilingual communication and entity-specific queries. This study addresses these limitations by developing a multimodal transformer framework capable of integrating video and textual inputs to generate semantically rich, entity-centric, and information-driven questions in a code-mixed Hindi-English format. Such a system is particularly significant for multilingual societies, offering applications in bilingual education, interactive learning platforms, conversational agents, and promoting cultural and linguistic relevance. To the best of our knowledge, there does not exist any large-scale Hindi-English (Hinglish) code-mixed dataset for video-based question generation. To address this limitation, we curated a subset of the TVQA dataset, and annotate it by bilingual experts, ensuring fluency, contextual appropriateness, and adherence to the code-mixed structure. Empirical evaluation shows that CoQuEST demonstrated competitive performance with metrics of RQUGE: 1.649, BLEU-1: 0.04, CIDEr: 0.29, METEOR: 0.20, Distinct-1: 0.96, Distinct-2: 0.99, ROUGE-L: 0.20, and BERT-Score F1: 0.88, validating its practical utility and effectiveness. We make the code and dataset publicly available.[1]

## 1 Introduction

The field of video question generation (VideoQG) has gained significant traction in recent years due to its potential applications in education, conversational AI, and interactive systems. With video content emerging as a dominant medium for information dissemination, learning, and entertainment, the need for tools that facilitate active engagement with such content has become increasingly apparent. VideoQG addresses this need by automatically generating semantically meaningful and contextually relevant questions from video content, transforming video consumption from a passive activity into an interactive experience Guo et al. (2021). Generation of code-mixed questions based on video poses unique challenges, particularly in multilingual societies. Code-mixing, the seamless blending of two or more languages within a sentence or conversation, is a natural linguistic phenomenon in such regions. For example, in India, where Hindi-English code-mixing is prevalent, phrases like "Madam, ये fabric प्योर cotton है। It's perfect for summer" *(Madam, this fabric is pure cotton. It's perfect for summer)*[2] are common. Code-mixed communication improves accessibility, cultural relevance and comfort, especially for bilingual users who naturally switch between languages Agarwal et al. (2021). However, most existing VideoQG systems are designed for monolingual applications, with English as the predominant language. This research aims to address this gap by developing a robust framework for generating semantically accurate, grammatically fluent, entity-centric, and contextually appropriate code-mixed Hindi-English questions from video content. To the best of our knowledge, there not exist any large-scale Hindi-English (Hinglish) code-mixed dataset for this objective, thus we extend a subset of the TVQA Lei et al. (2018) dataset by annotating with code-mix questions. Note, the entity-centric aspect arises from the underlying dataset question design, which targets entity-centric questions about characters and objects, and our annotation guidelines, which explicitly instruct annotators to

---

[1] https://anonymous.4open.science/r/CoQuEST-B28D/
[2] We translate the Code-mix text into English for the reader's convenience.

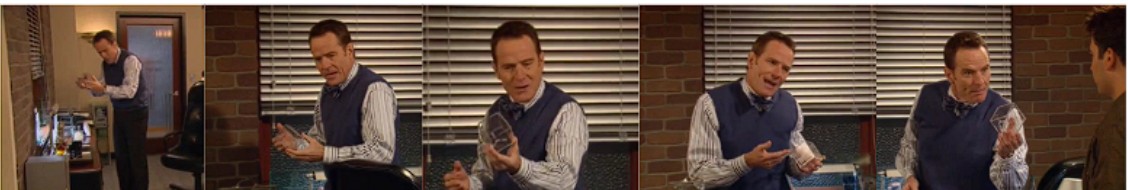

Figure 1: Example Code-Mixed VideoQG

preserve named entities and other salient content words when converting questions into code-mixed Hinglish. The motivation for this work lies in its diverse applications across multiple domains. In education Saputra (2023), code-mixed questions can aid students in bilingual regions to better comprehend tutorial videos. In corporate training and healthcare Dowlagar & Mamidi (2023), such systems ensure that instructional content is comprehensible and culturally relevant. Similarly, entertainment Jayashree et al. (2023) platforms and market research can engage bilingual audiences through code-mixed trivia, feedback analysis, and promotional content. We emphasize that MixTV-QA is a focused, domain-specific benchmark (sitcom videos), intended to enable controlled study of code-mixed multimodal question generation rather than to serve as a broad-coverage web-scale resource.

The primary contributions of this study include: **(1)** Developing a VideoQG model that integrates multimodal information from video and text inputs to generate code-mixed questions. This process included exploring two training pipelines for the architecture, as detailed in the Model Fine-Tuning Workflow (Section 4.1). We also evaluate our model on automated evaluation metrics such as BLEU Papineni et al. (2002), CIDEr Vedantam et al. (2015), METEOR Banerjee & Lavie (2005), Distinct Li et al. (2016), ROUGE Lin (2004), and BERT-Score Zhang et al. (2019). Recognizing the limitations of such metrics in capturing the quality of code-mix questions we also incorporate a reference-free metric, RQUGE Mohammadshahi et al. (2023). **(2)** Curating a novel dataset, MixTV-QA, consisting of code-mixed Hindi-English questions derived from diverse video content. **(3)** Addressing challenges in generating semantically coherent and grammatically correct code-mixed questions, ensuring their contextual relevance to bilingual users.

By achieving these objectives, this research aims to advance the field of multilingual question generation and contribute to the development of inclusive and accessible AI systems.

## 2 Related Work

### 2.1 Text-based Question Generation

Research in question generation (QG) has primarily focused on text-based approaches, as highlighted in works by Pan et al. (2019), Mitra et al. (2020; 2021) and Chatterjee et al. (2020). These studies have examined QG at various levels of granularity, such as document level Pan et al. (2020a); Yang et al. (2017); Tuan et al. (2020) paragraph level Du & Cardie (2018); Zhang et al. (2020), sentence level Ali et al. (2010) and

keyword level Pan et al. (2020b). This extensive exploration of text-based QG underscores the importance of leveraging contextual information at different hierarchical levels to generate meaningful questions.

## 2.2 Visual Question Generation

Visual Question Generation (VQG) was first introduced by Mostafazadeh et al. (2016) and further expanded upon by Patil & Patwardhan (2020) to generate questions from images. VQG questions are broadly categorized into three types. The first type, visually grounded questions, relies on the information directly available in the image itself, as demonstrated in studies by Antol et al. (2015); Krishna et al. (2017). The second type, commonsense-based questions, combines visual content with external commonsense knowledge, as explored by Wang et al. (2017b;a). The third type, world knowledge-based questions, integrates factual knowledge from external sources with visual information, as discussed in works by Shah et al. (2019); Penamakuri et al. (2023). Several methods have been employed for VQG, including encoder-decoder architectures Mostafazadeh et al. (2016); Zhang et al. (2017), compositional approaches Liu et al. (2018); Patro et al. (2018); Zhang et al. (2017), generative models Jain et al. (2017), reinforcement learning-based techniques Yang et al. (2018); Fan et al. (2018), and bilinear pooling models Fukui et al. (2016); Ben-Younes et al. (2017); Li et al. (2018). Recent research has also explored domain-specific adaptations of VQG, as highlighted by Mehta et al. (2024). This diverse array of approaches reflects the evolving complexity and scope of VQG, which extends beyond visual content to incorporate various external knowledge sources.

## 2.3 Code-Mix Video Question Generation

VideoQG has gained significant attention in recent years, focusing on the automatic generation of meaningful questions from video content. Early research in this domain primarily utilized visual features and transcripts to craft contextually relevant questions Yang et al. (2021); Su et al. (2021). This task shares similarities with Video Question Answering (VQA) but differs in its goal of generating questions rather than answering them. Efforts, such as MovieQA Tapaswi et al. (2016) have laid the groundwork for understanding multimodal content by creating benchmarks for reasoning over video-based narratives. Other works, like TGIF-QA Jang et al. (2017) and NewsVideoQA Jahagirdar et al. (2023), further explore temporal reasoning and domain-specific question generation.

Existing datasets in video question generation, such as SQuAD Rajpurkar (2016) and NEWSKVQA Gupta & Gupta (2022), focus on either textual comprehension or domain-specific tasks like news. These resources are valuable for advancing VQG and VQA research, but they do not address the situation where speakers switch between two or more languages within a single sentence. While MovieQA Tapaswi et al. (2016), SportsQA Li et al. (2024), and VQA-RAD Lau et al. (2018) provide questions on visual narratives, it, too, assumes monolingual inputs, failing to address the multilingual and code-mixed needs of diverse linguistic communities.

Code-Mixed Video Question Generation (Code-Mixed VideoQG) presents the unique challenge of generating questions from video content in code-mixed languages, such as Hinglish (Hindi-English), which is a prevalent linguistic phenomenon in multilingual communities. Although some research has explored code-mixed Visual Question Answering (Visual QA) Gupta et al. (2020), there remains a significant void in the form of dedicated datasets and methodologies tailored for Code-Mixed VideoQG. This gap underscores the critical need for advancing research to develop datasets and models capable of effectively handling code-mixing. For example, consider the task illustrated in Figure 1, where a question is generated based on a video clip in code-mixed format.

# 3 Dataset Creation

## 3.1 Data Curation and Preprocessing

We introduce **MixTV-QA**, a Hindi-English code-mixed QA dataset, built upon a subset of the publicly available TVQA dataset Lei et al. (2018). We selected 3,089 video clips from TVQA's six sitcoms (e.g., *Friends, The Big Bang Theory*). We manually refined the existing English questions into fluent code-

mixed (Hinglish) questions. Annotators, proficient in both languages, ensured grammatical fluency and semantic accuracy. This approach allowed us to leverage TVQA's high-quality video clips and transcripts while focusing our effort on creating high-quality code-mixed questions. The detailed steps are discussed in Appendix C.

## 3.2 Data Annotation

Two bilingual annotators translated the English questions from our TVQA Lei et al. (2018) subset into Hindi-English code-mixed questions following roughly 60-40 ratio (Hindi-English), designed to reflect natural usage by native Hindi speakers. Adhering to detailed guidelines (Refer Section C), they prioritized grammatical fluency, contextual relevance, and a conversational tone. Named entities were retained in English for clarity. Each question-answer pair was verified for its alignment with the corresponding video content to ensure contextual accuracy. Note: The 60–40 Hindi–English ratio is a dataset-level design target, not a hard per-sentence constraint. Annotators were instructed to produce questions where Hindi tokens form the majority but with substantial English content (roughly 60-40 when averaged across the corpus), rather than enforcing a strict exact ratio for each question. We verify the resulting degree of mixing via the standard metric, Code-Mixing Index Das & Gambäck (2014) (0.39), which indicates a medium level of code-mixing.

## 3.3 Quality Assurance and Validation

To ensure the quality of **MixTV-QA**, we implemented a validation strategy combining automated metrics and human evaluation. We measure agreement through an independent quality audit: each annotator rates the other's outputs on a small rubric (Relevance, Similarity, Fluency, Grammatical Correctness) using a 1–3 scale. Cohen's $\kappa$ scores indicate excellent inter-annotator agreement across all rubric criteria: Relevance (0.93), Similarity (0.95), Fluency (0.92), and Grammatical Correctness (0.90). See Table 6 for representative examples. We perform a sanity check using the ROUGE similarity score Lin (2004) to ensure consistency between annotators. Additionally, we compared the annotated questions with ChatGPT-generated outputs using BLEU, ROUGE, METEOR, and BERTScore, providing a quantitative assessment of lexical and semantic alignment. We analyzed the linguistic complexity of the code-mixed questions using the Code Mixing Index (CMI) Gambäck (2014) and Complexity Factor (CF) Ghosh et al. (2017) to gain insights into the degree of language mixing and linguistic diversity. Independent human evaluations, conducted by bilingual language experts distinct from the annotators, verified adherence to the prescribed 60-40 Hindi-English ratio, grammatical correctness, fluency, and contextual relevance. Deviations identified during this process were corrected to ensure high-quality annotations. Through these combined methods, we ensured that the MixTV-QA dataset, consisting of 3,089 code-mixed video-question pairs, meets a high-quality standard for Code-mixed VideoQG tasks. Additional details are discussed in Section C.

## 4 Methodology of CoQuEST

Given a video, at inference time, our proposed system generates code-mixed, video-based questions by integrating multimodal inputs, specifically video frames ($V$) and their corresponding transcript ($T$). The system employs a pipeline of modules, beginning with the extraction of embeddings from the video frames and transcripts, followed by fusion and generation of code-mixed questions. Formally, given the input video frames $V = \{v_1, v_2, \ldots, v_n\}$ and the corresponding transcript $T = \{t_1, t_2, \ldots, t_m\}$, the task is to learn a function

$$Q = q(V, T; \phi)$$

where $Q$ represents the set of code-mixed questions and $\phi$ denotes the model parameters. We illustrate the proposed method in Figure 2.

## 4.1 Code-Mix Question Generator

Our generator is designed to produce code-mixed questions (60% Hindi - 40% English) by leveraging video frames and their corresponding transcripts. From the dataset, 2,503 clips were used for training, with 20%

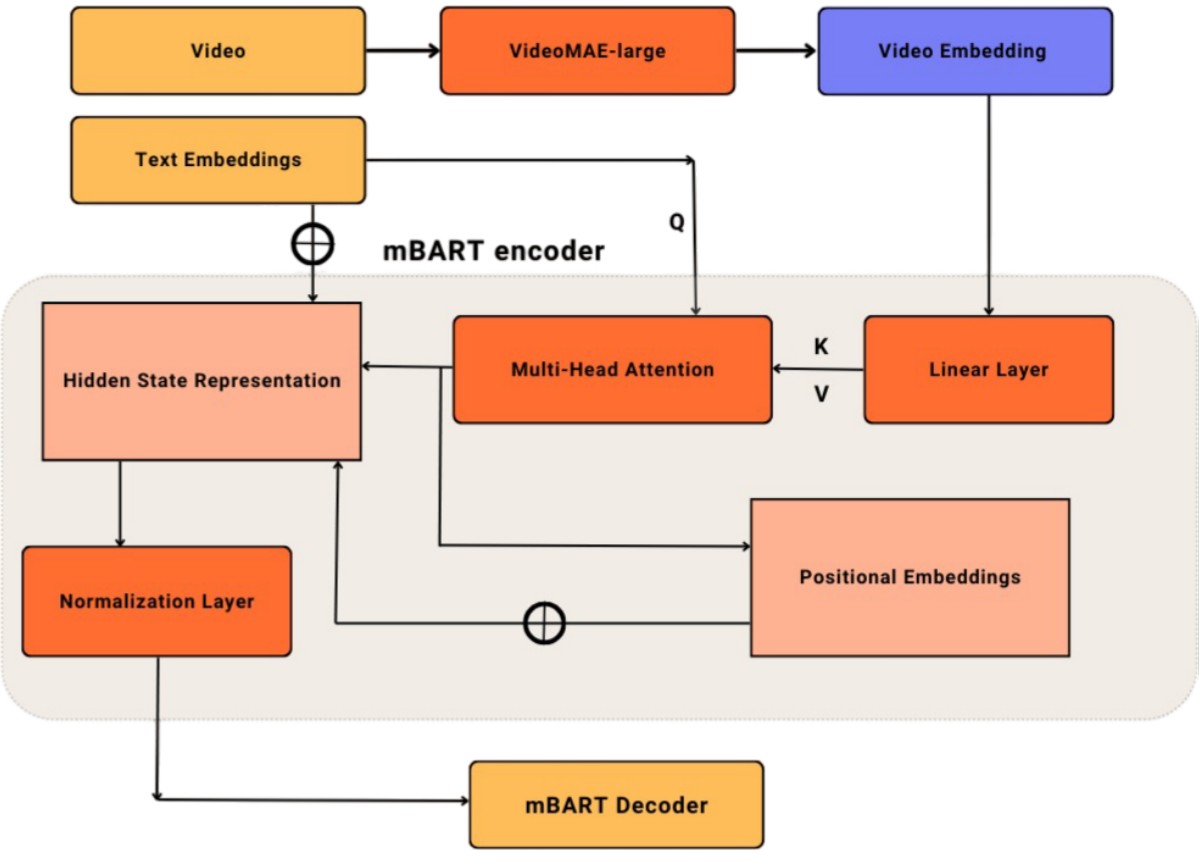

Figure 2: CoQuEST Architecture. **Takeaway:** *The hybrid architecture illustrates the information flow.*

of the dataset allocated for validation during training and the remaining designated as the test set for inference. The generator effectively captures the multimodal nature of the input, ensuring both linguistic and contextual fluency in the generated questions. A comparative analysis of the average word count for annotated and model-generated questions is provided in Table 4, highlighting the differences across question types and generation methods.

**Input Embeddings:** Our model integrates video and text modalities to generate code-mixed questions. We extract video embeddings using the VideoMAE-Large model Tong et al. (2022). For raw videos, we sample 16 frames uniformly, resize them to 224 x 224, and pre-process them into a tensor of dimension $\mathbb{R}^{16,3,224,224}$, where 16 denotes the number of frames, 3 corresponds to the color channels, and 224 x 224 represents the spatial resolution. Pre-extracted frames from TVQA were used directly. Text embeddings are derived from video transcripts using the mBART encoder Liu et al. (2020). We fuse modalities using a multi-head cross-attention mechanism across all mBART encoder layers. The video embeddings serve as the continuous key and value, while the text representation serves as the query. This allows the model to attend to relevant video context at every stage of encoding. The output of the final encoder layer, which contains the fused video-text representation, is passed to the mBART decoder. The decoder generates the code-mixed question in an auto-regressive manner. Our experiments found that this iterative, all-layer fusion strategy outperformed single-layer integration, underscoring the benefit of progressive refinement for multimodal alignment. The detailed pre-processing steps are discussed in Appendix D.1.

**Model Fine-Tuning Workflow:** We fine-tuned the encoder-decoder model using a typical cross-entropy loss, suitable for sequence generation tasks like question generation. Initially, a simple BART model proved

| Type | Model | RQUGE | BLEU-1 | CIDEr | ME-TEOR | Distinct-1 | Distinct-2 | ROUGE-L | BERT-Score F1 |
|------|-------|-------|--------|-------|---------|------------|------------|---------|---------------|
| Fine-tuned | Phukan et al. (2024) | 1.200 | 0.0003 | 0.00 | 0.01 | 1.61 | 2.22 | 0.00 | 0.77 |
| Fine-tuned | Asapu et al. (2025) | 1.262 | 0.0001 | 6.81 | 0.01 | 2.74 | 3.35 | 0.001 | 0.81 |
| Few-shot | Gala et al. (2024) | 1.209 | 0.0010 | 0.01 | 0.03 | 0.63 | 0.67 | 0.01 | 0.85 |
| Fine-tuned | DeepSeek-R1-Distill-Qwen-1.5B DeepSeek-AI (2025) | 1.200 | 0.0021 | 0.01 | 0.01 | 0.47 | 0.81 | 0.004 | 0.71 |
| Fine-tuned | Qwen2.5-7B Yang et al. (2024) | 1.225 | 0.0270 | 0.3 | 0.03 | 0.47 | 0.81 | 0.04 | 0.75 |
| Fine-tuned | Dabre et al. (2021) | 1.258 | 0.0239 | 0.21 | 0.18 | 0.87 | 0.92 | 0.20 | 0.88 |
| Zero-shot | VideoLLaVa-7B Lin et al. (2024) | 1.227 | 0.0016 | 0.01 | 0.04 | 1.46 | 1.77 | 0.01 | 0.83 |
| Zero-shot | Gemini 1.5 Flash Team et al. (2023) | 1.229 | 0.0057 | 0.01 | 0.08 | 1.74 | 2.12 | 0.04 | 0.84 |
| Zero-shot | Phi-4 Abdin et al. (2024) | 1.220 | 0.0041 | 0.00 | 0.03 | 1.64 | 2.10 | 0.03 | 0.84 |
| Zero-shot | SmolVLM Marafioti et al. (2025) | 1.218 | 0.0011 | 0.01 | 0.03 | 0.69 | 0.74 | 0.01 | 0.85 |
| Zero-shot | GPT-4o OpenAI (2024) | 1.231 | 0.0065 | 0.00 | 0.15 | 2.74 | 3.44 | 0.10 | 0.85 |
| Fine-tuned | **CoQuEST\*1**$(T, V, V\_D)$ *(ours)* | **1.325** | **0.0352** | **0.24** | **0.18** | **0.94** | **0.99** | **0.20** | **0.89** |
| Fine-tuned | **CoQuEST\*2**$(T)$ *(ours)* | 1.263 | **0.0185** | **0.14** | **0.13** | **0.70** | **0.68** | **0.14** | **0.88** |
| Fine-tuned | **CoQuEST\*2**$(T, V\_D)$ *(ours)* | 1.321 | **0.0183** | **0.13** | **0.12** | **0.77** | **0.77** | **0.12** | **0.86** |
| Fine-tuned | **CoQuEST\*2**$(T, V, V\_D)$ *(ours)* | 1.649 | **0.0450** | **0.29** | **0.20** | **0.96** | **0.99** | **0.20** | **0.88** |

Table 1: Evaluation Metrics for Different Models: $T$ = Transcript, $V\_D$ = Video Description, $V$ = Video Embedding. The table compares CoQuEST variants against other baseline models using RQUGE, BLEU-1, CIDEr, METEOR, Distinct-1 and 2, ROUGE-L (F1), and BERTScore F1. (i) *CoQuEST\*1* applies two-stage fine-tuning. (ii) *CoQuEST\*2* uses single-stage fine-tuning directly on code-mixed data. BLEU-1 values for CoQuEST\*2 are statistically significant over Dabre et al. (2021) (two-tailed t-test, $p = 4.71 \times 10^{-11} < 0.05$) signifying that our proposed architecture indeed helps with code-mixed VideoQG. ***Takeaway***: *Both CoQuEST variants surpass prior systems; CoQuEST\*2 with transcript + video embeddings + video descriptions yields the highest BLEU-1, while CoQuEST\*1 attains competitive ROUGE/BERTScore. Ablations show that dropping visual inputs hurts performance, evidencing the value of cross-modal fusion; semantic metrics (BERTScore) better capture quality than n-gram overlap alone.*

inadequate for handling the multilingual setup, particularly with code-mixed Hindi-English text. To address this, we opted for mBART, leveraging its robust multilingual capabilities. The fine-tuning process involved two distinct stages, forming the architecture we refer to as CoQuEST\*1. In the first stage, mBART was fine-tuned with Hindi labels, where the source language was set to English and the target language to Hindi. The mBART-large-50 weights were configured to perform this English-to-Hindi translation task. Video and text embeddings served as inputs, while corresponding Hindi questions acted as labels. During this stage, the model was optimized using the cross-entropy loss function to effectively align the English-to-Hindi translations.

In the second stage, the weights from the first stage were used to fine-tune the model for generating code-mixed questions. This stage formed the basis of CoQuEST\*2, where the annotations consisted of a mix of Hindi and English text, and no explicit source or target language parameters were specified. Instead, the model was trained to directly generate code-mixed questions, leveraging the pretrained English-to-Hindi translation knowledge to specialize in this task. Additionally, we experimented with a direct fine-tuning approach, training the model to generate code-mixed questions directly, without an intermediate English-to-Hindi translation stage. Although CoQuEST\*1's two-stage approach proved particularly effective with limited data, enabling the model to build foundational multilingual translation knowledge before specializing in code-mixed question generation, CoQuEST\*2 (the direct fine-tuning approach) demonstrated comparable

| Video Id | Gold Questions | CoQuEST Questions | EC |
|---|---|---|---|
| friends, s08e05, seg02, clip, 19 | जब चैंडलर ने Capricorn के बारे में कुछ कहा, तो मोनिका ने क्या पकड़ा? *(Eng:When Chandler said something about Capricorn, what did Monica catch?)* | जब Chandler ने Chandler को coffee दिया, तो Chandler ने Chandler से क्या कहा?*(Eng:When Chandler gave coffee to Chandler, what did Chandler say to Chandler?)* | RER |
| friends, s06e11, seg02, clip, 17 | जब Janine और Joey apartment में enter करते हैं, तो Janine क्या कहती है?*(Eng:When Janine and Joey enter the apartment, what does Janine say?)* | जब Joey Chandler से बात कर रहा होता है, तब Monica क्या कर रहा होता है?*(Eng:When Joey is talking to Chandler, what is Monica doing?)* | FI |
| friends, s02e16, seg02, clip, 19 | कौन कहता है कि वह Joey को याद करेगा क्योंकि वह uptown जा ने के बाद hall के उस पार नहीं रहेगा?*(Eng:Who says they will miss Joey because he won't be across the hall after moving uptown?)* | जब Chandler Monica से बात कर रहा होता है, तब Monica कहाँ होता है?*(Eng:When Chandler is talking to Monica, where is Monica?)* | GI |

Table 2: Examples of gold questions annotated by human annotators and corresponding questions generated by the CoQuEST model. EC (Error categories) : RER (Repetition Error), FI (Factual Inconsistency), GI (Grammatical errors). ***Takeaway***: *The examples illustrate the three dominant error types in our outputs. These are standard NLG pitfalls and are more pronounced in code-mixed settings, underscoring the need for code-mixed grammar correction.*

performance as the dataset size increased during later stages of dataset curation. A detailed discussion of these findings is presented in Section 5.2.

# 5  Experimental Setup and Results

## 5.1  Fine-tuning Results

During the fine-tuning step, we utilized input IDs and attention masks derived from concatenated prompts and transcripts representing the questions, while video embeddings were incorporated as a separate input entity into our modified model, CoQuEST. The model was trained on Tesla V100-PCIE GPUs, each with 32 GB of memory, over a total duration of 16.5 hours. The training process was conducted using a learning rate of 1e-5, with a batch size of 2 per GPU and gradient accumulation steps set to 2. The training was performed for 8 epochs for the CoQuEST*1 pipeline and 6 epochs for the CoQuEST*2 pipeline. Both pipelines followed the same configuration, including a weight decay of 0.01 and 10 warmup steps to stabilize learning. Mixed-precision training with FP16 was utilized for compatibility with CUDA 11.2, enhancing computational efficiency. Performance was monitored through an evaluation strategy that ran every 100 steps, with the best model selected based on the lowest evaluation loss. A save strategy was employed to retain only the two most recent models, and the maximum gradient norm was clipped to 1.0 to ensure stable optimization. Additionally, unnecessary columns were removed to optimize memory usage, ensuring an efficient and effective fine-tuning process.

## 5.2  Code-Mixed Question Generation Results

To evaluate the performance of our proposed CoQuEST models in generating code-mixed video questions, we conducted a comprehensive evaluation on a test set comprising 586 video clips and their corresponding transcripts. The evaluation was based on widely used metrics, including RQUGE, BLEU-1, CIDEr, METEOR, Distinct-1, Distinct-2, ROUGE-L, and BERT-Score F1, to assess syntactic accuracy, semantic alignment, and diversity. CoQuEST models were compared against several baselines, each tested using specific strategies tailored to its architecture and compatibility with the task. Detailed descriptions of these strategies are provided in Section E.

Phukan et al. (2024) was fine-tuned on our training set and then evaluated on the test set. IndicBART (Dabre et al. (2021)) was fine-tuned using video descriptions (generated by Qwen2-VL-7B Wang et al. (2024)) and transcripts, followed by inference on the test set. For Asapu et al. (2025), we fine-tuned the model on its original dataset using the official training script and then conducted inference on our test set using the fine-tuned weights. Airavata Gala et al. (2024) was tested using a few-shot approach, leveraging video descriptions and transcripts. Zero-shot evaluations were conducted for several models: VideoLLaVa-7B and

| Model | Rel | Fl | Cor |
|-------|-----|-----|-----|
| GPT-4o OpenAI (2024) | 1 | 2 | 2 |
| Phukan et al. (2024) | 1 | 1 | 1 |
| **CoQuEST*1***ours* | 2 | 2 | 2 |
| **CoQuEST*2***ours* | 2 | 3 | 2 |

Table 3: Human evaluation results. Here, Rel = Relevance, Fl = Fluency, and Cor = Correctness. ***Take-away***: *Both CoQuEST variants surpass prior systems; CoQuEST*2 achieves the highest scores, while CoQuEST*1 attains competitive results.*

Gemini 1.5 Flash used videos and transcripts as input, while Phi-4, SmolVLM, and GPT-4o relied on video descriptions and transcripts, with SmolVLM additionally tested on video data. The results, summarized in Table 1, demonstrate the superior performance of CoQuEST models over baselines. CoQuEST*1, using a two-stage fine-tuning approach, achieved BLEU-1 and CIDEr scores of 0.0352 and 0.24, respectively, while CoQuEST*2, employing single-stage fine-tuning directly on code-mixed data, scored 0.0450 and 0.29, respectively. CoQuEST models consistently outperformed baseline approaches across metrics like Distinct-1 and BERT-Score F1, highlighting their ability to generate diverse and semantically aligned questions. Baseline models exhibited varied performance. Phukan et al. (2024) struggled with low BLEU-1 (0.0003) and CIDEr (0.00) scores, reflecting challenges in handling code-mixed data. Asapu et al. (2025) performed similarly, with BLEU-1 and CIDEr scores of 0.0001 and 6.81, respectively. Few-shot evaluation of Gala et al. (2024) showed slight improvements, achieving CIDEr 0.01 but minimal diversity and semantic alignment. Models like VideoLLaVa-7B and Gemini 1.5 Flash demonstrated moderate performance, achieving BLEU-1 scores of 0.0016 and 0.0057, respectively, but struggled in diversity metrics. Dabre et al. (2021), fine-tuned on video descriptions and transcripts, showed stronger performance, with BLEU-1 and CIDEr scores of 0.0239 and 0.21, respectively, and better diversity compared to other baselines. Advanced baselines like SmolVLM, Phi-4, and GPT-4o exhibited similar limitations despite leveraging multiple inputs.

The low BLEU-1 and CIDEr scores across all models, including CoQuEST, can be attributed to the inherent variability in video question answering tasks. Videos often evoke multiple valid questions, reducing lexical overlap between generated and ground-truth questions. Mathematically, BLEU emphasizes n-gram precision, while CIDEr computes cosine similarity of weighted n-gram counts, both penalizing low overlaps. However, CoQuEST models achieved high Distinct-1 and Distinct-2 scores (0.94-0.99) and the highest BERT-Score F1 values (0.88-0.89), underscoring their ability to generate diverse and semantically aligned questions despite the variability in phrasing. To further investigate the impact of input modalities, we conducted an ablation study using CoQuEST*2. Specifically, we evaluated its performance using only transcripts (T), and a combination of transcripts and video descriptions (T,V_D). Interestingly, adding auxiliary video descriptions (V_D) does not consistently improve performance over Transcript-only or Transcript+Video settings in our ablations (Table [X]). We hypothesize that this is due to noise and bias introduced by automatically generated descriptions: (i) they may omit the entities relevant to the question, (ii) they may introduce hallucinated or overly generic content, and (iii) their style may be mismatched to the code-mixed question space. As a result, V_D can act as a distractor signal during generation. This finding is practically important: it suggests that more text is not always better in multimodal QG, and that description quality control (filtering, confidence scoring, or contrastive grounding) is likely required for V_D to be beneficial. We leave robust description calibration as future work.

Overall, our proposed fine-tuning strategies proved highly effective, with CoQuEST*2 excelling in scalability for larger datasets, while CoQuEST*1 demonstrated strong performance for limited training data. These findings, along with the detailed evaluation strategies and ablation study, highlight CoQuEST's robustness as a solution for Code-mixed VideoQG.

## 5.3 Qualitative Analysis

**Human Evaluation** In this Code-mixed VideoQG task, automated evaluation metrics alone are insufficient to fully assess the quality of generated questions. Video content often supports multiple valid question scenarios, extending beyond the scope of the predefined gold questions. Human evaluation plays a crucial role in capturing nuanced aspects of question quality, especially in the context of code-mixed languages. The

human evaluation of generated questions was conducted based on three core criteria: relevance, fluency, and correctness/accuracy, each capturing a specific aspect of question quality and alignment with the video's content. Detailed scoring guidelines for these criteria are provided in the Section F.

Relevance measured how closely the generated question aligned with the video's main content. A high score (3) indicated that the question directly addressed a core topic or key point in the video, while a low score (1) suggested that the question was off-topic or irrelevant. Fluency assessed the grammatical correctness, coherence, and naturalness of the generated questions. A high score reflected well-formed, grammatically correct sentences with a natural conversational tone, whereas a low score indicated awkward phrasing or significant grammatical errors. Correctness/accuracy evaluated the factual alignment of the question with the video's content. High-scoring questions were entirely accurate and free of misinformation, whereas low scores were assigned to questions containing factual errors or misrepresentations of the video content. For instance, consider the generated question: Wilson के office में घुसने से पहले House क्या कर रहा था? *(What was House doing before entering Wilson's office?)*. This generated question received scores of 3, 3 and 3 for relevance, fluency, and correctness/accuracy. It was directly aligned with the video's content, grammatically well-formed, and factually accurate. The corresponding gold question for this example was: Cameron ने House के sarcastic सवाल का जवाब देते हुए क्या कहा कि उसे उसका बॉस किसने बनाया था? *(What did Cameron say in response to House's sarcastic question about who made him her boss?)*. This gold question set a benchmark for evaluating the generated questions against human expectations of quality. CoQuEST*1 and CoQuEST*2 consistently outperformed baseline models, such as GPT-4o OpenAI (2024), by achieving higher scores across all these parameters. Despite these strengths, evaluators identified occasional shortcomings, such as awkward phrasing, indicating areas for further refinement. The human evaluation scores (Table 3) demonstrate the significant advancements made by CoQuEST*1 and CoQuEST*2 in producing contextually relevant, fluent, and accurate code-mixed questions.

**Error Analysis** The generated questions exhibited several common errors that impacted their overall quality and alignment with the video content. A frequent issue was the inclusion of irrelevant content (IR), which often arose from inconsistencies between the video and its textual descriptions, resulting in questions mentioning characters or topics unrelated to the video. Another recurring error was the repetition of words or phrases (RER), where redundancy reduced the clarity and effectiveness of the questions. Fluency issues (FI) further affected the questions, where the outputs lacked a natural conversational tone, making them awkward or stilted. Grammatical errors (GI) were also observed, particularly in the Hindi-English code-mixed output. These errors typically stemmed from the grammar of Hindi (the primary input language) influencing sentence structure. For example, issues related to word order or the omission of auxiliary verbs were more common due to the syntactic differences between Hindi and English. Additionally, some code-mixed outputs showed an imbalance in linguistic integration, with one language dominating excessively, resulting in questions that failed to achieve a true blend of Hindi and English. To better understand these issues, we conducted a human evaluation on a subset of inference results, analyzed by independent evaluators. The analysis revealed that 46.67% of the evaluated subset exhibited no errors. However, irrelevant content (IR) constituted 36.67% of the identified issues, often due to mismatches between video content and textual descriptions. Repetition of words or phrases (RER) accounted for 10.00%, reducing question clarity. Fluency issues (FI), observed in 3.33% of cases, resulted in awkward or unnatural phrasing. Grammatical issues (GI), also at 3.33%, were influenced by Hindi's syntactic structure, particularly in code-mixed outputs. These findings highlight key areas requiring improvement, such as handling primary-language grammar (Hindi), improving fluency, and ensuring a balanced bilingual approach. A summary of these errors, along with illustrative examples, can be found in Table 2.

## 6 Conclusion

This work addresses the significant research gap in Code-mixed Video Question Generation (VideoQG). We make four key contributions: (1) introducing the task itself, (2) creating and releasing MixTV-QA, the first dataset for Hindi-English code-mixed VideoQG, (3) proposing CoQuEST, a novel model that effectively fuses video and text modalities via iterative cross-attention, and (4) establishing a strong benchmark for this task. Empirical results show that CoQuEST outperforms comparable baselines on standard metrics (BLEU, ROUGE, METEOR), validating the importance of deep, multi-layer multimodal fusion. By providing a

dedicated dataset, a state-of-the-art model, and comprehensive evaluations, this work lays a foundation for future research in multilingual and multimodal question generation, thereby paving the way for more inclusive AI systems.

## 7 Limitations

While our research has made significant strides in video-based question generation for code-mixed languages, there are several limitations that highlight areas for future improvement.

**Dataset Scope and Scale:** The MixTV-QA dataset, while a crucial first step, is limited in size and domain diversity. Its focus on scripted sitcoms does not fully capture the spontaneity and linguistic variation found in real-world videos (e.g., vlogs, interviews). Future work could expand to larger, more varied datasets encompassing multiple genres and spontaneous dialogues.

**Script Representation:** Our dataset uses the Devanagari script for Hindi. However, Romanized Hindi ("Hinglish") is prevalent in informal digital communication. Extending the model to handle this Romanized code-mixing would greatly enhance its practical applicability.

**Question Complexity:** Our model excels at generating factual questions but struggles with complex, reasoning-based, or multi-hop questions. Enhancing the model's capability for deeper semantic understanding and reasoning remains an important challenge.

Ultimately, this work serves as a foundational step in video-based question generation for code-mixed languages. By addressing both the quality of code-mixing and the unique structural characteristics of video content, we establish a framework that can be expanded to accommodate more diverse linguistic and conversational contexts.

## 8 Ethical Considerations

We recognize that multimodal VideoQG systems carry inherent ethical risks, starting with the possibility of bias amplification. Since our model is trained in part on TVQA data, any societal stereotypes or demographic skew in that dataset may be reflected or even exacerbated in the generated questions. Closely tied to this is the risk of unequal code-mixing behavior; the model might overuse one language over another in contexts where that language usage reflects systemic bias. We also acknowledge privacy concerns, especially if such a system were deployed on personal or sensitive videos. Although our current work utilizes only publicly available data, real-world deployments would require robust privacy safeguards (e.g., anonymization, on-device inference). Because generative models can produce plausible but incorrect or misleading content, we emphasize that responsible deployment must include continuous monitoring, bias audits, and human oversight to detect and correct unfair, unclear, or harmful outputs.

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

## A  Overview

To complement the findings and experiments described in the main paper, this supplementary material includes the following:

- A detailed description of the annotation process of the data set, including the guidelines followed to annotate **MixTV-QA**, linguistic patterns, and code-mixing strategies. This section also presents a few representative samples from the dataset.

- An in-depth explanation of the fine-tuning process for the CoQuEST pipeline proposed in the main paper, covering architectural design choices and training strategies.

- Step-by-step descriptions of how we evaluated baseline models mentioned in the main paper, including strategies, prompts, and configurations used for running inference on these models.

| Question Type | Average Word Count |
|---|---|
| Hindi Annotated | 25.19 |
| English Annotated | 13.38 |
| Code-mixed Annotated | 21.36 |
| GPT-4o Code-mixed | 64.86 |
| **CoQuEST*1** Code-mixed | 19.09 |
| **CoQuEST*2** Code-mixed | 18.83 |

Table 4: Average Word Count for Questions

- Human evaluation examples, highlighting generated outputs, qualitative assessments, and feedback on their linguistic and contextual relevance.

For a full understanding of the context and research objectives, readers are encouraged to refer to the main paper.

## B Rationale for Code-Mixed Video Question Generation Benchmark

This section elaborates on the motivation for introducing a benchmark that evaluates models on the joint task of video grounding and natural code-mixed question generation, as addressed in our work.

**Motivation and Real-World Alignment:** Our primary goal is to address a prevalent but under-studied mode of user interaction in multilingual regions. In contexts like India (with high populations of Hindi–English speakers), dialogue around video content often occurs naturally in code-mixed language (e.g., Hinglish), not in strictly monolingual forms. While the technical components of video question generation (VideoQG) and code-mixing are individually studied, no existing benchmark evaluates a model's integrated ability to perform visually grounded code-mixed generation that mirrors this organic speech pattern. Limitations of Pipeline Approaches. A straightforward alternative, generating a monolingual question and then applying machine translation (MT) or a code-mixing module, suffers from critical shortcomings Zhang et al. (2025); Sheth et al. (2025). Such pipelines optimize stages in isolation, leading to the loss of key properties we aim to retain:

- Natural Mixing Distribution: Human code-mixing follows patterns (e.g., Hindi grammar with embedded English named entities) that are often not captured by off-the-shelf MT/code-mixing systems, which may produce entirely monolingual outputs or unnatural, token-level alternations Zhang et al. (2025).

- Semantic Precision: The translation/code-mixing step can distort or incorrectly translate named entities and other video-grounded lexical items, breaking the critical link between the visual content and the generated text Dowlagar & Mamidi (2022).

- Stylistic Consistency: Pipeline outputs frequently lack the stylistic consistency of how bilingual speakers actually converse, sounding either overly formal or artificially mixed Kodali et al. (2025).

**Application Scope and Complementary Baselines:** We do not propose code-mixed output as universally superior to monolingual text. Instead, we target the substantial real-world ecosystems where code-mixed interfaces are already the default for bilingual users, such as Hinglish search queries, subtitles, and chatbots on Indian OTT platforms. We consider simpler monolingual-generation-plus-translation pipelines as important but complementary baselines. Their limitations in joint optimization and explicit control highlight the need for an end-to-end approach. Our model is trained directly on human-authored, video-grounded code-mixed questions, enabling it to learn contextual switching behavior, such as retaining English proper nouns and show-specific terms while using Hindi for grammatical structure, conditioned on the multimodal input.

## C Dataset Creation

### C.1 Annotation Overview

The annotation process was a critical component of the **MixTV-QA** dataset curation pipeline, ensuring the generation of high-quality code-mixed questions from the pre-existing English questions in the TVQA dataset. The process involved adapting questions to a code-mixed Hindi-English format while retaining their semantic and contextual accuracy. A standardized set of guidelines was established to assist annotators in maintaining consistency across annotations. These guidelines focused on ensuring fluency, grammatical correctness, and contextual relevance while adhering to the prescribed 60-40 Hindi-English ratio. Named entities, such as names and locations, were preserved in English for clarity, while transitions between the languages were crafted to reflect natural code-mixed speech patterns.

We hired two bilingual annotators proficient in Hindi and English to adapt the questions for the **MixTV-QA** dataset. During the dataset annotation phase, all annotators performed an intermediate step of first translating the English TVQA labels to pure Hindi in Devanagari script. This Devanagari script was later codemixed to Hindi and English. We use these intermediate Hindi Devanagari annotations to train the first stage of CoQuEST*1. The role of the annotators were instrumental in ensuring that the final dataset adhered to the highest standards of linguistic and contextual accuracy. They were compensated according to the institute's pay scale for their efforts, emphasizing the value of their expertise.

| Show Name | GOLD Question | CoQuEST Generated Question |
|---|---|---|
| Friends | जब Phoebe उसे गा रही थी, तब Rachel क्या कर रही थी? (*Eng: When Phoebe is singing, what is Rachel doing?*) | जब Chandler apartment में enter करता है, तो उसके हाथ में क्या होता है? (*Eng: When Chandler enters the apartment, what does he have in his hand?*) |
| The Big Bang Theory | जब Amy और Penny अचानक आती हैं, तो Sheldon, Howard, Raj और Bernadette अपनी beads पर क्या पहन रहे है? (*Eng: When Amy and Penny suddenly arrive, what are Sheldon, Howard, Raj, and Bernadette wearing on their beads?*) | जब Sheldon कहती है "I'm sipping Kool-Aid through a Red Vine..." तो Howard क्या कहता है? (*Eng: When Sheldon says, "I'm sipping Kool-Aid through a Red Vine…", what does Howard say?*) |

Table 5: Examples of Code-mixed questions annotated and generated by CoQuEST for MixTV-QA

### C.2 Annotation Guidelines

To ensure the quality and consistency of the **MixTV-QA** dataset, a detailed set of annotation guidelines was established. These guidelines were designed to facilitate the generation of semantically accurate and grammatically fluent code-mixed questions while adhering to the contextual nuances of bilingual Hindi-English users. The annotation process focused solely on question generation, following Hindi grammar rules as the primary framework.

## Key Components of the Guidelines

- **Language Usage:**
  - **Code-Mixed Format:** All questions were required to be in a Hindi-English code-mixed format, ensuring natural language flow.
    **Example:** "यह चैनल number क्या है?"
    Annotators ensured fluency and coherence by maintaining a conversational tone, making the questions accessible and engaging for bilingual users.

- **Fluency and Grammar:**
  - **Fluency:** Questions were crafted to flow naturally and be easily comprehensible, reflecting everyday conversational speech.
    **Example:** "तुम्हारा favourite sports कौन सा है?"
  - **Grammatical Accuracy:** Since Hindi was the primary grammatical framework, all annotations adhered to Hindi grammar rules while seamlessly integrating English lexical items.

**Correct:** "नीतीश Kumar और चंद्रबाबू Naidu की राजनीतिक स्थिति क्या है?"
**Incorrect:** "नीतीश Kumar और चंद्रबाबू Naidu की राजनीत स्थिति क्या है?"

- **Contextual Relevance:**

  - **Hallucination Handling:** Questions strictly avoided introducing any information not supported by the video or its context.
    **Correct:** "इस clip में character ने क्या कहा?"
    **Incorrect:** "इस video में दिखाया गया मैच किसने जीता?"
    Annotators ensured that every question was logically relevant to the video's content and contextually appropriate.

- **Sentence Rewriting:**

  - Annotators were permitted to rephrase questions to improve clarity and readability, provided the original meaning was preserved.
    **Original:** "Meeting का time क्या है?"
    **Rewritten:** "Meeting कब है?"

- **Error Correction:**

  - **Error Types:** Annotators identified and corrected typographical, grammatical, and semantic errors.
    **Typo:** "य channel number क्या है?" → "यह channel number क्या है?"
    **Grammatical:** "यह channels number क्या है?" → "यह channel number क्या है?"

### C.3 Quality Assurance and Validation

To ensure the quality and consistency of the annotations, we implemented the following measures:

- **Sanity Check:** To ensure consistency and maintain the quality of the dataset, we evaluated the similarity between annotations using ROUGE-L scores. For this, we selected a subset of overlapping questions generated by both annotators based on the same videos. The average ROUGE-L similarity between the annotations was found to be 0.8788, validating the alignment and coherence of the annotated questions. This process highlights the reliability of the dataset annotations for code-mixed question generation tasks.

- **Independent Human Evaluation**

  To validate the quality of the code-mixed annotations, independent bilingual language experts evaluated the questions generated based on four key dimensions. The evaluations ensured that the code-mixed questions adhered to the expected standards of naturalness, relevance, similarity, and grammatical correctness. Below are the evaluation criteria, (Refer Table 6 for examples):

  - **Fluency:** Evaluates whether the code-mixing is natural and fluent, reflecting how bilingual speakers actually communicate.
    * **Score 1:** The code-mixing is awkward or forced. Example: What is आपका favourite color? (unnatural blending of English and Hindi).
    * **Score 2:** The code-mixing is somewhat natural but has minor issues. Example: आपका favourite color क्या है? (acceptable but not optimal flow).
    * **Score 3:** The code-mixing is seamless and natural. Example: तुम्हारा favourite color कौन सा है? (perfectly fluent and conversational).
  - **Relevance:** Assesses whether the code-mixed question aligns with the intent and content of the original English question.
    * **Score 1:** The question is off-topic or deviates significantly. Example: किस देश में यह event हुआ? (irrelevant to the video's content).

* **Score 2:** The question is somewhat related but not entirely focused. Example: इस event के participants कौन हैं? (partially relevant but lacks precision).
* **Score 3:** The question directly aligns with the video's main content. Example: यह event किस जगह पर हुआ? (fully relevant and contextually appropriate).

- **Similarity to Original Question:** Measures how closely the code-mixed question resembles the structure and meaning of the original English question, while being appropriately code-mixed.
  * **Score 1:** The question significantly diverges in structure or meaning. Example: Original: *"What is the name of the speaker?"* Generated: कौन इस event को host कर रहा है? (meaning is changed).
  * **Score 2:** The question maintains partial similarity but includes unnecessary alterations. Example: Original: "What is the name of the speaker?" Generated: Speaker का नाम क्या हो सकता है?(similar but slightly modified).
  * **Score 3:** The question closely aligns with the original. Example: Original: What is the name of the speaker? Annotated: Speaker का नाम क्या है?

- **Grammatical Correctness:** Ensures that the question follows proper grammar, with a focus on Hindi grammar rules as the primary language in code-mixed annotations.
  * **Score 1:** Contains major grammatical issues. Example: Speaker का नाम क्या हैं? (incorrect use of plural verb form with singular subject).
  * **Score 2:** Contains minor grammatical issues but remains understandable. Example:Speaker का नाम क्या था?(acceptable but contextually incorrect tense).
  * **Score 3:** Grammatically correct and contextually appropriate. Example: Speaker का नाम क्या है? (correct and adheres to Hindi grammar rules).

| Original Question | Code-Mixed Question | Relevance | Similarity | Fluency | Grammatical Correctness |
|---|---|---|---|---|---|
| Why Derek is sad when talking to Thatcher? | डेरिक Thatcher से बात करते समय sad क्यों है? | 3 | 3 | 3 | 3 |
| What did Jake say Elena needed to do before the surgery? | Jake ने Elena से क्या कहा कि उसे surgery से पहले क्या करना चाहिए? | 3 | 3 | 3 | 3 |
| What did Izzie see after the MRI came out clean? | MRI normal आने के बाद Izzie ने क्या देखा? | 3 | 3 | 3 | 3 |
| Where are Cristina and Alex when they are talking? | जब Cristina और Alex बात कर रहे होते हैं, तो वे कहाँ होते हैं? | 3 | 3 | 3 | 3 |
| What body part is bandaged up on the patient when McdDreamy is in the room with the patient? | जब McDreamy patient के साथ room में होता है, तो patient के body का कौन सा part bandaged होता है? | 2 | 3 | 3 | 3 |

Table 6: Examples of Evaluated Code-Mixed Questions with Scores

## Comparison with ChatGPT Generated Questions

To evaluate the quality of the annotated questions, we compared them with questions generated by ChatGPT using a specific prompt. The prompt used for ChatGPT to generate code-mixed questions was as follows:

"Rewrite the given questions in a code-mixed version (60-40 Hindi (Devanagari script) - English) respectively. Focus on Named Entity Recognition (NER) such as places and names, ensuring they are properly code-mixed. Strictly follow these rules as the speaker is a native Hindi speaker: In the code-mixed version, the starting word or phrase cannot be in English. Maintain a 60-40 ratio (Hindi-English) in each line. Ensure that NERs (places, names, etc.) are in English in the code-mixed version."

Using this prompt, ChatGPT was tasked with generating code-mixed versions of questions. The evaluation of these questions yielded the following results:

- **BLEU-1:** 0.35

- **CIDEr:** 3.04

- **METEOR:** 0.72

- **Distinct-1:** 1.00

- **Distinct-2:** 1.00

- **ROUGE-L (F1 score):** 0.60

- **BERTScore (F1):** 0.95

The high values for **BERTScore F1** and **CIDEr** indicate that the annotated questions align well with the ChatGPT-generated questions in terms of semantic and contextual relevance. Note, we compute BERTScore F1 using the *summ_eval* toolkit, specifically the *BertScoreMetric* wrapper *(from summ_eval.bert_score_metric import BertScoreMetric)*. We use the default English configuration provided by summ_eval, which corresponds to: *lang='en', model_type='bert-base-uncased', num_layers=8, verbose=False, idf=False, nthreads=4, batch_size=64, rescale_with_baseline=False.*

Furthermore, the **BLEU** and **ROUGE-L** scores show strong lexical similarity between the two sets. The distinctiveness metrics (**Distinct-1** and **Distinct-2**) suggest that both the annotated and ChatGPT-generated questions exhibit a rich variety of expressions. These results validate the robustness of the annotation process and the overall quality of the **MixTV-QA** dataset.

## D CoQuEST Pipeline Details

### D.1 Input Embeddings

In our system, video embeddings are a fundamental component for generating code-mixed video-based questions. To extract these embeddings, we utilized the VideoMAE-Large model Tong et al. (2022), which produces embeddings of size 1024. For raw video files, we employed the PyAV module integrated into the VideoMAE pipeline to extract frames. Specifically, we selected 16 evenly spaced frames from each video, ensuring adequate coverage of the visual content. These frames were resized to 224 x 224 pixels, converted to RGB format and pre-processed to meet the input requirements of the VideoMAE model. The processed frames were stacked and reshaped to form a tensor of dimension $\mathbb{R}^{16,3,224,224}$, where 16 denotes the number of frames, 3 corresponds to the color channels, and 224 x 224 represents the spatial resolution. Video frames from the TVQA subset, already pre-extracted, were directly utilized, eliminating the need for dynamic extraction. To maintain consistency in input representation, all frames were processed uniformly, regardless of their source.

The video embeddings generated by the VideoMAE-Large model are integrated with text embeddings derived from the transcript of the respective videos. These textual embeddings are created using the mBART Liu et al. (2020) encoder. To effectively fuse the video and text modalities, we employ a multi-head cross-attention mechanism. In this setup, the video embeddings act as the key and value, while the text embeddings serve as the query. This enables the system to identify and prioritize the video segments most relevant to the textual content, ensuring a meaningful alignment between the two modalities. After the initial fusion, the combined text embeddings are enhanced with positional embeddings to retain sequence order information. These processed embeddings are then passed through the first encoder layer of mBART. In subsequent encoder layers, the hidden representations from the preceding layer are iteratively refined using multi-head cross-attention with the video embeddings. During this iterative process, the video embeddings continue to function as the key and value, while the outputs of the previous encoder layer serve as the query. This step-by-step refinement allows the model to progressively align and integrate visual and textual features, capturing

both temporal dependencies in the video and semantic nuances in the text.The output from the final mBART encoder layer represents the fully integrated video-text information, which is then forwarded to the mBART decoder. The decoder generates the code-mixed questions in an auto-regressive manner, producing one token at a time until the entire question is constructed. Our experiments showed that applying cross-attention between video embeddings and textual representations across all encoder layers consistently outperformed strategies where the integration occurred only in the final encoder layer, highlighting the importance of iterative refinement for effective multi-modal fusion.

### D.2 CoQuEST*1: Two-Stage Fine-Tuning Approach

The CoQuEST*1 pipeline follows a two-stage fine-tuning methodology to leverage intermediate translation knowledge for effective code-mixed question generation.

- **Stage 1: Hindi Question Generation**
  - **Prompt Used:**

    ```
    Generate a Hindi question from the given video and transcript.
    Transcript: {transcript}.
    ```
  - **Inputs:** Video embeddings (extracted using VideoMAE-Large) and transcripts were provided to the model. Target labels consisted of Hindi questions.
  - **Fine-Tuning Objective:** Train a customized encoder-decoder architecture (using mBART as the base) capable of processing video modalities and text inputs to align English text with Hindi outputs using a cross-entropy loss.

- **Stage 2: Code-Mixed Question Generation**
  - **Prompt Used:**

    ```
    Generate a Code-mix question from the given video and transcript (60
    percent Hindi and 40 percent English). Transcript: {transcript}.
    ```
  - **Inputs:** Video embeddings and transcripts were used again, with the weights fine-tuned in Stage 1 as the initialization. Target labels consisted of code-mixed questions.
  - **Fine-Tuning Objective:** Specialize the model for code-mixed question generation while leveraging pretrained translation knowledge from Stage 1 using mBART as the base architecture.

### D.3 CoQuEST*2: Direct Fine-Tuning Approach

The CoQuEST*2 pipeline adopts a single-stage fine-tuning approach, directly training the model for code-mixed question generation.

- Prompt Used:

  ```
  Generate a Code-mix question from the given video and transcript (60
  percent Hindi and 40 percent English). Transcript: {transcript}.
  ```

- Inputs: Video embeddings and transcripts were provided, and target labels were code-mixed questions.

- Fine-Tuning Objective: Directly optimize the model for code-mixed question generation without intermediate translation stages.

### D.4 Comparison of Approaches

CoQuEST*1 demonstrated superior performance for smaller datasets due to its staged approach, enabling the model to build foundational multilingual translation knowledge before specializing in code-mixed generation. However, CoQuEST*2 showed comparable or better performance as dataset size increased, suggesting the direct fine-tuning approach benefits from larger-scale data availability.

**Performance Scores:**

- **CoQuEST*1 (ours):**
  - BLEU-1: **0.03**
  - CIDEr: **0.24**
  - METEOR: **0.18**
  - Distinct-1: **0.94**
  - Distinct-2: **0.99**
  - ROUGE-L: **0.20**
  - BERT-Score F1: **0.89**

- **CoQuEST*2 (ours):**
  - BLEU-1: **0.04**
  - CIDEr: **0.29**
  - METEOR: **0.20**
  - Distinct-1: **0.96**
  - Distinct-2: **0.99**
  - ROUGE-L: **0.20**
  - BERT-Score F1: **0.88**

Further details on the performance evaluation of these pipelines against other baselines are discussed in the main text. Note: Our choice of mBART-large was driven by the following considerations: **(1)** mBART has strong support for Hindi and Devanagari script, which is essential for our code-mixed setting, and integrates cleanly with our encoder–decoder architecture and VideoMAE-based visual encoder. **(2)**The focus of this work is not on competing with the latest very large LMs, but on studying a modular, multimodal code-mixed VQG pipeline (CoQuEST). We therefore opted for a stable, well-understood backbone and then compared this pipeline against a range of strong baselines (IndicBART, Airavata, recent VLMs such as VideoLLaVA, SmolVLM, Phi-4, GPT-4o, and Gemini 1.5 Flash). **(3)** mBART-large (600M) offers a good trade-off between capacity and resource requirements. It allows other researchers to reproduce our results on a single-GPU setup, whereas fully fine-tuning very large models such as Aya-101 (13B)[3] is substantially more demanding.

## E   Baseline Evaluation Strategies

To comprehensively evaluate the performance of our proposed CoQuEST models, we employed several baseline models. The following outlines the strategies used to fine-tune, test, or evaluate each baseline:

- **ECIS-VQG: Generation of Entity-centric Information-seeking Questions from Videos**
  - Used the architecture proposed in the original paper for question generation tasks, which supports both textual and video inputs.
  - Fine-tuned directly on the training set of **MixTV-QA** with code-mixed questions as the target outputs.
  - Video embeddings, generated using the VideoMAE-Large model, were passed as input alongside the transcript.
  - The following prompt was used during fine-tuning:
    ```
    Generate a Code-mix question from the given video and transcript (60
    percent Hindi and 40 percent English). Transcript: [Transcript].
    ```
  - Evaluated on the test set of **MixTV-QA**, focusing on generating code-mixed questions.
  - Results: Achieved BLEU-1 of 0.00, CIDEr of 0.001, METEOR of 0.012, Distinct-1 of 1.62, Distinct-2 of 2.22, ROUGE-L of 0.002, and BERT-Score F1 of 0.77.

---

[3]https://huggingface.co/CohereLabs/aya-101

- **IndicBART: A Pre-trained Model for Natural Language Generation of Indic Languages.**
  - Fine-tuned using video descriptions (generated by Qwen2-VL-7B) and transcripts as inputs.
  - The following prompt was used during fine-tuning:

    ```
    Generate a Code-mix question (60 percent Hindi and 40 percent
    English) from the given video and transcript. Transcript: [Transcript]
    Description: [video_description].
    ```
  - Fine-tuning employed the `Seq2SeqTrainer` class from the HuggingFace Transformers library, with these training parameters:
    * Learning rate: $3 \times 10^{-5}$
    * Batch size: 4 (both training and evaluation)
    * Number of epochs: 7
    * Weight decay: 0.01 .
  - Evaluated on the test set with both input modalities (transcripts and video descriptions).
  - Results: Achieved BLEU-1 of 0.02, CIDEr of 0.21, METEOR of 0.18, Distinct-1 of 0.87, Distinct-2 of 0.92, ROUGE-L of 0.20, and BERT-Score F1 of 0.88.

- **Bridging Laughter Across Languages: Generation of Hindi-English Code-mixed Puns**
  - Fine-tuned the model from scratch by replicating the original training script described in the paper.
  - The original fine-tuned weights were unavailable, so we re-trained the model on its original dataset and subsequently used those weights to perform inference on our test set.
  - During inference, the following prompt was used:

    ```
    Generate a code-mixed (60% Hindi, 40% English) question based on:
    Description:['video_description'] and Transcript: ['Transcript'].
    ```
  - Evaluated on the test set of **MixTV-QA** using code-mixed data.
  - Results: Achieved BLEU-1 of 0.00016, CIDEr of 0.0, METEOR of 6.81, Distinct-1 of 2.74, Distinct-2 of 3.35, ROUGE-L of 0.001, and BERT-Score F1 of 0.81.

- **DeepSeek-R1-Distill-Qwen-1.5B**
  - Fine-tuned using video descriptions (generated by Qwen2-VL-7B) and transcripts as inputs.
  - The following prompt was used during fine-tuning:

    ```
    Generate a Code-mix question (60 percent Hindi and 40 percent
    English) from the given video and transcript. Transcript: [Transcript]
    Description: [video_description].
    ```
  - Fine-tuned directly on the training set of MixTV-QA with code-mixed questions as the target outputs.
  - Evaluated on the test set with both input modalities (transcripts and video descriptions).
  - Results: Achieved BLEU-1 of 0.0021, CIDEr of 0.01, METEOR of 0.01, Distinct-1 of 0.47, Distinct-2 of 0.81, ROUGE-L of 0.004, and BERT-Score F1 of 0.71.

- **Qwen2.5-7B** [4]
  - Fine-tuned using video descriptions (generated by Qwen2-VL-7B) and transcripts as inputs.
  - The following prompt was used during fine-tuning:

    ```
    Generate a Code-mix question (60 percent Hindi and 40 percent
    English) from the given video and transcript. Transcript: [Transcript]
    Description: [video_description].
    ```
  - Fine-tuned directly on the training set of MixTV-QA with code-mixed questions as the target outputs.

---

[4]`https://huggingface.co/Qwen/Qwen2.5-7B`

- Evaluated on the test set with both input modalities (transcripts and video descriptions).
- Results: Achieved BLEU-1 of 0.0270, CIDEr of 0.3, METEOR of 0.03, Distinct-1 of 0.47, Distinct-2 of 0.81, ROUGE-L of 0.04, and BERT-Score F1 of 0.75.

- **Airavata: Introducing Hindi Instruction-tuned LLM**

  - Tested using a few-shot prompting approach with video descriptions and transcripts as inputs.
  - Few-shot prompts included examples demonstrating how to generate code-mixed questions with 60% Hindi in Devanagari script and 40% English, based on the given inputs.
  - Prompt Template:

    ```
    Below are examples of how to generate code-mixed questions (60% Hindi
    in Devanagari script and 40% English) based on video descriptions and
    transcripts.
    Example 1:       Video Description: The video shows a boy flying a kite on
    a sunny day in a park. Other children are playing in the background.
    Transcript: Today, we are learning how wind helps a kite fly. The boy
    runs to catch the wind, and the kite rises higher into the sky.  Generate
    a code-mixed question based on the video (Use 60% Hindi in Devanagari
    script and 40% English): हवा का क्या role होता है kite उड़ाने में?
    Example 2:       Video Description: The video is a classroom setting where
    a teacher is explaining water cycle using a diagram. The students are
    watching attentively.       Transcript: Water evaporates from the surface,
    forms clouds, and returns as rain. This cycle is essential for life on
    Earth.   Generate a code-mixed question based on the video (Use 60% Hindi
    in Devanagari script and 40% English): पानी का cycle कैसे complete होता है Earth
    पे?
    Now generate a question for the following:            Video Description:
    {description}   Transcript: {transcript}   Generate a code-mixed question
    based on the video (Use 60% Hindi in Devanagari script and 40% English):
    ```
  - Evaluated on the **MixTV-QA** test set.
  - Results: Achieved BLEU-1 of 0.00, CIDEr of 0.01, METEOR of 0.04, Distinct-1 of 0.25, Distinct-2 of 0.33, ROUGE-L of 0.01, and BERT-Score F1 of 0.85.

- **VideoLLaVa-7B and Gemini 1.5 Flash**

  - Zero-shot evaluations performed using video and transcript inputs.

- **Phi-4, SmolVLM, and GPT-4o**

  - Zero-shot evaluations were conducted using video descriptions and transcripts as inputs.
  - SmolVLM was additionally tested using video data as an input modality.
  - Prompt Template:

    ```
    Generate a Code-mix question in 60% Hindi and 40% English based on the
    transcript and description.       Use Devanagari script for Hindi and the
    English alphabet for English.    Transcript: '{transcript}'. Description:
    '{description}'.
    ```
  - Evaluated on the **MixTV-QA** test set.
  - Results: Achieved BLEU-1, CIDEr, METEOR, Distinct-1, Distinct-2, ROUGE-L, and BERT-Score F1 scores as specified in the main text for each model.

## F  Human Evaluation Guidelines for CoQuEST Generated Questions

Human evaluation was conducted by two Hindi–English bilingual annotators, both native Hindi speakers fluent in English and familiar with code-mixed Hinglish in Devanagari and Latin script. Annotators were recruited via internal calls and compensated according to the institute's standard

| Model | What it is | Why chosen | Inputs | Training on MixTV-QA |
|---|---|---|---|---|
| ECIS-VQG Phukan et al. (2024) | Entity-centric VideoQG model (ECIS-VQG) with joint video+text encoding | Prior entity-focused VideoQG architecture, adapted to code-mixed output | T, V, V_D | Fine-tuned on MixTV-QA code-mixed questions |
| Asapu et al. (2025) | Generates Hindi-English code-mixed puns by identifying phonetically similar word pairs across two languages | Strong Hindi / English generator easily adaptable to Hinglish | T, V_D | Fine-tuned on its original dataset, then zero-shot transfer to MixTV-QA |
| Airavata Gala et al. (2024) | Instruction-tuned LLM for Hindi and 22 other Indic languages | Representative Hindi NLG model | T, V_D | Few-shot prompting on MixTV-QA (no fine-tuning) |
| IndicBART Dabre et al. (2021) | Pre-trained encoder–decoder LM for Indic languages | Strong Hindi / Indic generator easily adaptable to Hinglish | T, V_D | Fine-tuned on MixTV-QA code-mixed questions |
| VideoLLaVA-7B Lin et al. (2023) | Multimodal LLM (vision+language) | Strong open-source VLM baseline | T, V, V_D | Zero-shot prompting on MixTV-QA |
| Gemini 1.5 Flash Team et al. (2023) | Proprietary multimodal LLM | State-of-the-art commercial VLM baseline | T, V, V_D | Zero-shot prompting on MixTV-QA |
| Phi-4 Abdin et al. (2024) | Compact text-only LLM | Strong small LM baseline for text-only code-mixed generation | T, V_D | Zero-shot prompting on MixTV-QA |
| SmolVLM Marafioti et al. (2025) | Small vision–language model | Lightweight VLM designed for efficient multimodal inference | T, V, V_D | Zero-shot prompting on MixTV-QA |
| GPT-4o OpenAI (2024) | Large multimodal LLM | High-capacity commercial baseline for Hinglish generation | T, V_D | Zero-shot prompting on MixTV-QA |
| CoQuEST*1 (ours) | Our two-stage pipeline (EN->HI then HI->code-mix) | Tests staged translation + code-mix specialisation | T, V, V_D | Fine-tuned (Stage 1 on Hindi translations, Stage 2 on MixTV-QA) |
| CoQuEST*2 (ours) | Our single-stage code-mixed fine-tuning | Directly optimises for code-mixed VideoQG | T, V, V_D | Fine-tuned on MixTV-QA |

Table 7: Details of the baselines implemented

hourly pay scale for research assistants. Before rating, they completed a training session on a set of 20 examples, with detailed guidelines and discussion to calibrate scoring for Relevance, Fluency, and Correctness. The inter-annotator agreement (kappa score) on the 100 random samples are mentioned in Table 9:

**Relevance**

**Definition:** Measures how well the generated question aligns with the core content of the video.

**Guidelines:**

- **Score 1:** The question is completely off-topic or irrelevant to the video content. For example, in a video about renewable energy, a question like "आसमान का रंग क्या है?" (What color is the sky?) would be marked as irrelevant.
- **Score 2:** The question is somewhat related to the video content but not central to the main topic. For instance, a question like "मनुष्यों का pollution में कितना योगदान है?" (How do humans contribute to pollution?) in a video about renewable energy might be considered somewhat relevant but not the primary focus.
- **Score 3:** The question is directly related to the core content of the video and addresses a key point. For example, "सोलर energy के क्या मुख्य फायदे हैं?" (What are the key benefits of solar energy?) in a video focused on renewable energy would receive a high relevance score.

**Fluency**

**Definition:** Assesses the grammatical correctness, coherence, and naturalness of the question.

**Guidelines:**

- **Score 1:** Not fluent / hard to understand. The question is fragmented or unreadable due to broken mixing or structure. Example: का आसमान है रंग क्या? (sky is color what?)
- **Score 2:** Mostly fluent with minor awkwardness. Reads fine overall, but has small rough edges (slight unnatural switch, minor phrasing awkwardness). For example, मनुष्यों का pollution में how much योगदान है? is slightly awkward while मनुष्यों का pollution में कितना योगदान है? (How do humans contribute to pollution?) is more sensible.
- **Score 3:** Highly fluent and natural. Smooth, conversational, and code-switching feels organic. For example, सोलर energy के क्या मुख्य फायदे हैं? (What are the key benefits of solar energy?).

**Correctness/Accuracy**

**Definition:** Evaluates if the generated question reflects the accurate information presented in the video.

**Guidelines:**

- **Score 1:** The question contains factual errors or misrepresents the video content. For example, सोलर ऊर्जा कैसे non-renewable resource है? (How is solar energy a non-renewable resource?) would be inaccurate since solar energy is renewable.
- **Score 2:** The question is mostly accurate but may contain small factual errors or misunderstandings. An example could be, सोलर ऊर्जा electricity generate करते समय wind कैसे use होता है? (How does wind generate electricity through solar energy?), which mixes up solar and wind energy.
- **Score 3:** The question is factually correct and accurately represents the video content. For example, सोलर ऊर्जा electricity कैसे generate करती है? (How does solar energy generate electricity?) is both accurate and relevant.

For examples of Human Evaluation score of CoQuEST generated code-mixed question refer to Table 8.

| Gold Question | CoQuEST Generated Question | Relevance | Fluency | Correctness/Accuracy |
|---|---|---|---|---|
| Cameron ने House के sarcastic सवाल का जवाब देते हुए क्या कहा कि उसे उसका बॉस किसने बनाया था? | Wilson के office में घुसने से पहले House क्या कर रहा था? | 3 | 3 | 3 |
| Chandler को surprise क्यों हुआ जब Kathy ने अपने date की तरफ इशारा किया? | जब Joey Chandler से बात कर रही होती है, तो वह कहाँ बैठी होती है? | 3 | 3 | 3 |
| Rachel ने Chandler के "paisan" के बारे में कुछ कहने के बाद क्या किया? | जब Chandler Joey से बात कर रहा होता है, तो वह कहाँ खड़ा होता है? | 3 | 2 | 3 |
| Izzie कहती है कि उसे लगता है कि Hannah को सुअर पसंद हैं क्योंकि उसकी माँ ने उसे "Charlotte's Web" पढ़ा था? | जब Izzie कहती है "She likes pigs", तो Derek कहती है, "She collects them..."? | 3 | 3 | 3 |

Table 8: Example of Human Evaluation score of CoQuEST

| Models | Relevance | Fluency | Correctness |
|---|---|---|---|
| GPT-4o OpenAI (2024) | 0.90 | 0.76 | 0.79 |
| ECIS-VQG Phukan et al. (2024) | 0.92 | 0.68 | 0.77 |
| CoQuEST*1 (ours) | 0.91 | 0.79 | 0.76 |
| CoQuEST*2 (ours) | 0.93 | 0.72 | 0.79 |

Table 9: Inter-annotator agreement (kappa score) on 100 random samples

## Code and Data Availability

For access to the code, dataset, and additional resources related to this research, please refer to the project's GitHub repository: `https://anonymous.4open.science/r/CoQuEST-B28D/`. The repository includes the implementation of the CoQuEST pipeline, sample of the **MixTV-QA** dataset, as well as detailed instructions for usage, replication of experiments, and further contributions.

## Conclusion

This supplementary material has provided detailed information to further clarify the methodologies and evaluations discussed in the main paper, "CoQuEST: Entity-Focused Code-Mixed Question Generation for Entertainment Videos". It has outlined the annotation process for the **MixTV-QA** dataset, the fine-tuning procedures for the CoQuEST pipeline, evaluation strategies for baseline models, and examples of human evaluation to ensure a comprehensive understanding of our approach and its results. By offering a deeper understanding of the technical aspects, we aim to provide the necessary transparency and reproducibility for others in the research community to engage with and build upon our work. We believe this supplementary content strengthens the foundation laid by the main paper and offers valuable details that will assist researchers in the domain of multimodal processing and code-mixed question generation.

