# OpenReview forum: "CoQuEST: Entity-Focused Code-Mixed Question Generation for Entertainment Videos"
_TMLR — Rejected by TMLR_

### Review · Reviewer_z6pV · 2026-02-09

**Summary Of Contributions:**

The paper introduces MixTV-QA, a dataset for Hindi-English code-mixed Video Question Generation (VideoQG), consisting of 3,089 pairs derived from sitcom clips. It proposes CoQuEST, a multimodal transformer framework based on mBART that fuses video embeddings (via VideoMAE) and transcript text through iterative cross-attention. The authors evaluate several fine-tuning strategies and report improvements over various text-based and zero-shot VLM baseline

Key Strengths:

- Addresses a practical need for multilingual communities where code-mixing is the natural communication mode.

- The iterative cross-attention mechanism across all encoder layers shows a clear benefit over late-fusion approaches.

Key Weaknesses:

- Potential for confirmation bias in human evaluation due to overlapping roles of annotators and evaluators.

- Dataset Redundancy: The high overlap between human "Gold" questions and GPT-4o translations suggests the expert annotation may not offer significant value beyond automated translation of the source dataset

- Weak Baselines: Several baseline comparisons are hampered by using noisy text descriptions as a proxy for video content rather than direct video input

**Audience:**

No

**Audience Explanation:**

No, because the findings are insolid. While VideoQG in code-mixed settings is a relevant topic, the methodology relies on older backbones (mBART-large-50) and fails to provide a rigorous comparison against SOTA video-native models (or just VLMs like qwen3-VL). The evidence suggesting the dataset is nearly identical to automated translation further limits the potential impact for the TMLR audience.

**Claims And Evidence:**

No

**Claims Explanation:**

Redundancy of Manual Annotation: The paper reports a BERTScore F1 of 0.95 and CIDEr of 3.04 when comparing human annotations to GPT-4o outputs generated from the original English questions. This extremely high overlap suggests that the human "expert" annotation is primarily performing a translation task that current LLMs already handle with near-human parity. Without proving that human annotations capture unique, organic code-mixing nuances that LLMs miss, the necessity of the new dataset is unconvincing

Evaluator Independence: The paper states that human evaluation was conducted by "independent" experts, yet the supplementary material clarifies that these were the same "two Hindi-English bilingual annotators" who presumably helped create or curate the data. This lack of separation compromises the objectivity of the qualitative results.

Inconsistent Baseline Modalities: Many baselines (DeepSeek-R1-Distill-Qwen-1.5B) were restricted to textual video descriptions, which are generated by Qwen2-VL-7B. Comparing a model that has direct access to video embeddings against models forced to use flawed text proxies creates an unfair and unconvincing advantage for CoQuEST

**Requested Changes:**

Independent Qualitative Study: Perform a new human evaluation using participants who were not involved in the original data curation or annotation.

Distinguish "Gold" from GPT: Provide a detailed qualitative analysis or a "Turing Test" to prove that the human-annotated Hinglish questions are superior to or more "natural" than the GPT-4o translations. If the data is effectively a translation of TVQA, the authors must justify the manual effort.


Direct Video Baselines: Re-evaluate SOTA models like VideoLLaVA or Gemini 1.5 Flash using direct video input for all trials to ensure a fair comparison against the proposed multimodal fusion.

---

> ### Author Response · Authors · 2026-02-19
>
> > Potential for confirmation bias in human evaluation due to overlapping roles of annotators and evaluators.
> * We thank the reviewer for pointing out this ambiguity. We wish to clarify that dataset annotators did not participate in model output evaluation. Evaluation was performed by separate bilingual evaluators who were not involved in dataset creation, using the same scoring rubric. We will update the revised draft to make it clearer.
>
> > Dataset Redundancy
> * Thank you for pointing this out. This similarity is expected because both outputs are derived from the same underlying English question, and our GPT prompt explicitly enforces constraints (Section C). High BERTScore/CIDEr therefore indicates that GPT can preserve the semantic intent, not that the outputs are identical in code-mixing naturalness, style, or grounding discipline. Additionally, we observed that GPT generations tend to be longer and more verbose (avg length 44.86 vs 21.36 words for human code-mixed questions), suggesting a style mismatch for VideoQG where concise, naturally code-mixed questions are preferred. We will also incorporate a Blind preference (Turing-style) study in the revised draft, where independent bilingual raters will be shown Gold vs GPT rewrites (randomized, anonymized) and asked which is more natural/appropriate Hinglish for the given clip+transcript; we will report win-rate and agreement.
>
> > Weak Baselines
> * We wish to clarify that the suggested VideoLLaVA and Gemini 1.5 Flash are present and use video input (as mentioned in Sections 5.2 and E). Additionally, we report the results of ECIS-VQG (Phukan et al., 2024), which was fine-tuned on video inputs. We relied on video descriptions to compare against sota text only models like GPT-4o (api inference) and DeepSeek R1-Distill-Qwen1.5B
>
> > Backbone choice (mBART) and relevance to TMLR audience
> * We chose mBART-large-50 to ensure strong Hindi/Devanagari support and reproducibility on modest compute, and we already evaluated against several recent VLMs. That said, we agree the paper will be stronger with clearer positioning: the primary contribution is the first Hindi-English code-mixed VideoQG benchmark in this controlled domain and a transparent multimodal fusion baseline. We will tighten claims accordingly in the revised draft.

---

### Review · Reviewer_pPnV · 2026-03-05

**Summary Of Contributions:**

This paper studies the generation of code-mixed (hybrid language) questions from videos. It focuses on the case of generating Hindi-English mixed language, aka Hinglish, from entertainment videos. There are two main contributions: 1) a carefully curated dataset of videos to corresponding Hinglish questions, and 2) a proposed model design to perform the question generation task. In-distribution automatic evaluation and human evaluation showed the proposed model outperforms a variety of baselines, including multi-modal large language models like Gemini 1.5 Flash and GPT-4o.

The key strength of the paper is the special dataset curated for this task.

The key weakness is the narrow scope of the problem. While the significance and value could be clear in certain culture or commercial applications, the approach is less novel and the value to broader ML research is limited. This feels more like a nicely done public or commercial project than ML research.

**Additional Comments:**

The review does appreciate the author's effort to work in less studied languages and culturally meaningful tasks.

**Audience:**

No

**Audience Explanation:**

As I said above, the scope of the problem is too narrow and too close to a concrete application.

**Broader Impact Concerns:**

No ethical concern.

**Claims And Evidence:**

No

**Claims Explanation:**

It's a mixed case.

The quality of the dataset is validated by metrics like inter-annotator agreement and linguistic distribution analysis.

However, the conclusion of the proposed model architecture is better and of research value (aka generalization is expected) is weak, because of the following concerns:

- The evaluation is mostly in-distribution with the training data.
- The automatic metrics are measuring against a pre-defined set of questions, while the nature of the task allows different but still legit questions. The task don't have closed, precise answer. The paper also pointed this out.
- The human evaluation, designed to address the second concern, doesn't show the right level of confidence interval. Details like the number of examples evaluated is not provided clearly,  and the level of noise is not measured.
- Moreover, the results in Table 3 are confusing: are they the average value of multiple examples' scores? If so, why are they all perfect integers?

**Requested Changes:**

To make it more valuable to the broader ML community, it could be helpful to broaden the task to more language pairs, and more types of videos. It's also helpful to study the nature of code-mix language generation by using out-of-distribution dataset to understand the generalization process.

The rapid advance of general purpose multi-modal LLMs also calls for stronger baseline, e.g. fine-tuning or few-shot on the same task. That could be more generalizable or economical (in terms of method complexity) than building special purpose models.

The paper also needs to be more clear about the human evaluation metrics, addressing the details on confidence interval, sample size and how the scores in the table are calculated.

---

> ### Author Response · Authors · 2026-03-08
>
> >  narrow scope of the problem
> * The paper targets a broader methodological gap: code-mixed multimodal generation, where the output distribution (switching behavior, script mixing, entity retention) is not well-served by monolingual VideoQG benchmarks. Our contributions are (a) a reproducible benchmark for code-mixed VideoQG, and (b) a transparent multimodal fusion baseline that can be extended to other language pairs. The current draft explicitly frames MixTV-QA as a focused, controlled benchmark rather than a web-scale resource: “MixTV-QA is a focused, domain-specific benchmark (sitcom videos), intended to enable controlled study of code-mixed multimodal question generation rather than to serve as a broad-coverage web-scale resource.”
> * As per reviewers' suggestion, we will add a short discussion of how the annotation protocol and model design can be reused for other language pairs and domains (news, tutorials, vlogs), while keeping MixTV-QA as a controlled starting point.
>
> > The evaluation is mostly in-distribution with the training data.
> * The current evaluation uses the standard train/test split within the same sitcom distribution (test set size reported as 586 clips). We will add an out-of-distribution generalization study using show-level splits (e.g., hold out one show entirely during training and test on that unseen show). This evaluates cross-show generalization without collecting new data. We will also report performance by show/genre slice to quantify distribution sensitivity.
>
> >  The task don't have closed, precise answer. The paper also pointed this out
> * As noted in Section 1, we acknowledge the inherent limitations of automatic metrics in evaluating the quality of code-mixed questions, given that the task is open-ended and does not presuppose a single correct answer. To address this, we already include a reference-free metric, RQUGE, in our evaluation.
> * To further strengthen our assessment, we will incorporate a downstream task that evaluates question-answering utility. Specifically, we aim to investigate whether the hidden representations from CoQuEST*2 can support the generation of plausible answers when provided with the English question, along with the corresponding transcript and video embeddings. We evaluated CoQuEST*2 on the TVQA dataset as a question-answering task, thereby offering a more functional and task-oriented evaluation:
> |Model| Accuracy(%)|
> | ---- | ------------------- |
> |RUBi (Cadene et al. 2019)|64.75%|
> |STAGE(Lei et al. 2020)|66.92%|
> |TVQA (Lei et al. 2018)|67.70%|
> |CoQuEST*2|68.85%|
>
> > Details like the number of examples evaluated is not provided clearly
> * Section 3 and F provides a clear rubric and scoring definition (Relevance/Fluency/Correctness on a 1–3 scale), and the supplementary describes rater calibration (20 training examples) and inter-annotator agreement checks on 100 samples which is also the sample size of the human evaluation results mentioned in Table 3. Noise is measured via agreement: K ranges 0.68–0.93 across criteria/models on N=100, Table 9.
>
> > The results in Table 3 are confusing
> * Thank you for flagging this. Table 3 uses the 1–3 ordinal scale (1 = low, 3 = high), and the table currently presents rounded aggregate scores, which can be misinterpreted. We will replace Table 3 with unrounded means.
>
> >  it could be helpful to broaden the task to more language pairs, and more types of videos.
> * Thank you for pointing this out. To further strengthen our assessment, we will incorporate a downstream task that evaluates question-answering utility. And also add an out-of-distribution generalization study using show-level splits (e.g., hold out one show entirely during training and test on that unseen show)
>
> > The rapid advance of general purpose multi-modal LLMs also calls for stronger baseline, e.g. fine-tuning or few-shot on the same task.
> * We cover multiple model families in the draft. We can group baselines into three categories:
> * (a) Finetuned baselines on MixTV-QA (e.g., IndicBART, ECIS-VQG, Asapu et al.),
> * (b) Few-shot prompted LLMs (e.g., Airavata), and
> * (c) Zero-shot multimodal LLMs (e.g., GPT-4o, Gemini, VideoLLaVA, SmolVLM).
> * The Input modalities and training details of all these baselines are already discussed in section E in the appendix.

---

> > ### Comment · Reviewer_pPnV · 2026-03-15
> > **Thanks for the quick response. I'm still holding my general opinion.**
> >
> > Scope: I saw the author's effort to add more discussion about it and planned future work, but it won't change the scope that the evidence in the paper can support. It's still pretty narrow in the Hindi-English case in entertainment-type shows.
> >
> > In-distribution evaluation. Here I mean the type of language pair and video. Adding random split and show-level held-out won't change the concern on in-distribution data (i.e. the same narrow scope concern).
> >
> > Evaluation: the use of RQUGE (reference free metric) becomes the key focus here. The set up of RQUGE is non-trivial, e.g. what question-answering module are you using? Is there any other work that used it on multi-modal Q&A? The original RQUGE paper only showed results on text datasets.
> >
> > About the clarity of Table 3. I re-read the paper and can't find a clear statement that there are 100 samples in the dataset. One line in the appendix says that inter-annotator consistency is evaluated on 100 samples, but didn't say that the evaluation set is of size 100. We should not rely on appendix for any critical information either.
> >
> > Given the concerns above, I still hold my opinion. Thanks.

---

### Review · Reviewer_ygyz · 2026-03-09

**Summary Of Contributions:**

The paper introduces MixTV-QA which is modification of TVQA by changing the questions to code-mixed questions. The paper also introduces a code-mixed Video Question generation model.

Strengths:
- The paper gives every detail about annotation, and evaluation with examples, which would help future works.
- The paper clearly describes the model used for video Question generation. They also try all-layer fusion vs single-layer integration.
- The paper performs ablations between text transcripts, video descriptions and videos as inputs.
- The paper introduces two fine-tuning strategies and finds one better for smaller dataset and one for larger.

Weaknesses:
- The distinct-1 and distinct-2 measures show CoQuEST produce significantly less diverse questions than VideoLLaVa-7B, Gemini 1.5, Phi-4, GPT-4o.
- The human evaluation compares only against GPT-4o and Phukan et al. It is not clear what is the rationale behind using these two. Also the human evaluation lacks details regarding number of samples evaluated and if the reported results are average scores.
- Some of the examples show unnatural use of code-switching. For example, the example in the introduction, pure is written in the Devanagari (Hindi) script but it makes much more sense to have it in English since it is a English word. Similarly in Table 6, Derek is written in Hindi. I wonder if the constraint in getting close to 60/40 Hindi/English results in this. Also the rationale behind choosing 60/40 is not clearly explained with references.
- The rationale behind the setting is not clear. Why is it needed to generate code mixed questions from English videos? Wouldn't it make more sense to generate code-mixed questions from code-mixed videos since people consuming English media should prefer English questions and people consuming code mixed media or Hindi media might prefer code-mixed questions.

**Audience:**

No

**Audience Explanation:**

No, the rationale behind the setting is not well explained and the claims are not well supported.

**Claims And Evidence:**

No

**Claims Explanation:**

The claims are well supported by the automatic and the human evaluations. However, some claims regarding better diversity is not reported correctly as some baselines (VideoLLaVa-7B, Gemini 1.5, Phi-4, GPT-4o) have better diversity. But in the text it is claimed that CoQuEST has the best diversity. Similarly some CIDER scores are also higher for baselines (Qwen2.5-7B, Asapu et al). Same for ROGUE-L Dabre et al. has similar scores.

**Requested Changes:**

- Table 1 is misleading. CoQuEST does not have the best scores across all metrics but is always bolded. This should be corrected
- Provide rationale regarding the baselines chosen in the Human evaluation and details regarding number of samples evaluated and if the reported scores are average.
- Provide citations/rationale behind choosing the 60/40 split.
- Explain the rationale behind the setting of generating code-mixed questions from English videos. Wouldn't it make more sense to generate code-mixed questions from code-mixed videos since people consuming English media should prefer English questions and people consuming code mixed media or Hindi media might prefer code-mixed questions.

---

> ### Author Response · Authors · 2026-03-17
>
> > Human evaluation baselines and missing details
> * Our rationale for choosing Phukan et al. is that it is the closest supervised VideoQG baseline in our domain, and GPT-4o is a strong general-purpose LMM baseline widely used for multilingual generation. Section 3 and F provides a clear rubric and scoring definition (Relevance/Fluency/Correctness on a 1–3 scale), and the supplementary describes rater calibration (20 training examples) and inter-annotator agreement checks on 100 samples, which is also the sample size of the human evaluation results mentioned in Table 3 (results are mean scores over N). Noise is measured via agreement: K ranges 0.68–0.93 across criteria/models on N=100, Table 9.
>
> > Some of the examples show unnatural use of code-switching.
> * We acknowledge that certain examples may appear unnatura, such as rendering English tokens or named entities in Devanagari script. This arises from our annotation protocol, which permits script mixing and transliteration; the dataset preserves annotators' original choices, including instances where they opted for phonetic rendering in Hindi script. Such cases are not merely an artifact of the 60/40 constraint, but can also reflect colloquial Hinglish writing styles in which transliteration is commonplace.
> * That said, as per the reviewer’s suggestion, We will (i) add citations from code-mixing literature motivating a target mixing range (rather than a hard rule) and clarify that our guideline is a soft preference, not an enforced constraint; (ii) include a short analysis showing the distribution of mixing ratios across the dataset; (iii) explicitly state that named entities should remain in their original form unless commonly transliterated, and we will revise examples in the paper (Intro/Table 6) to avoid unintuitive transliterations and present more representative samples.
>
> > Rationale of the setting: why code-mixed questions for English videos?
> * We thank the reviewer for raising this important motivation question. Our goal is not to claim that English media implies English-only questions; rather, in many Hindi-dominant user populations, English entertainment content is consumed with Hindi/ Hinglish commentary and queries (e.g., bilingual viewers discussing English clips, subtitles, or dubbed contexts). In practice, Hinglish is frequently used as a query language even when the media source is English, because the user’s preference is code-mixed, not necessarily the video’s spoken language. MixTV-QA thus models cross-lingual / mixed-language query generation grounded in English content, which is relevant for multilingual retrieval, video assistants, and accessibility.
> * We will strengthen the introduction with this motivation (user-language != content-language), and we will add a discussion paragraph about how the protocol can extend to code-mixed videos as well (a natural future direction).
>
> > Table 1 formatting / bolding and metric claims
> * We will update Table 1 to bold only the best score per metric (or best among comparable modality settings), and use additional formatting (e.g., underlining) for second-best when helpful. We will also correct any statements that incorrectly claim best-in-column when baselines match or exceed our score (e.g., CIDEr/ROUGE-L cases noted by the reviewer).

---

### Decision · Action_Editor_sAnJ · 2026-04-13

**Recommendation:** Reject

**Audience:**

No

**Audience Explanation:**

The topic of code-mixed video question generation is meaningful. However, the paper focuses on a narrow task and setting, and does not convincingly establish broader insights, generalization, or a clearly distinct dataset contribution. The comparison with more recent multimodal models is also limited. As a result, the work is unlikely to be of sufficient interest to the broader TMLR audience.

**Claims And Evidence:**

No

**Claims Explanation:**

The paper includes both automatic and human evaluation, and demonstrates that the system can operate on the proposed benchmark. However, the evidence is not sufficiently convincing to support the main claims, for the following reasons:
* Mismatch between task structure and automatic metrics.
The task is inherently open-ended: a single video can support multiple valid questions. However, most reported metrics evaluate against a fixed reference question. This setup systematically underestimates valid alternative outputs and makes it difficult to interpret reported improvements as meaningful gains in question quality.
* Human evaluation does not fully resolve this limitation.
While intended to address the above issue, the human evaluation is not presented with sufficient statistical clarity. Key details such as sample size, aggregation method, and variability are not clearly reported, and the use of rounded integer scores further reduces interpretability. As a result, it is difficult to assess the reliability and significance of the qualitative findings.
* The distinct value of the dataset is not clearly established.
MixTV-QA is derived from a subset of TVQA by manually rewriting the original English questions into a code-mixed form. The reported high similarity between human annotations and GPT-based rewrites raises a central question: whether the dataset captures properties of natural code-mixing that cannot already be reproduced by strong LLMs. This question is not convincingly answered in the current submission.
* Model-level claims are not sufficiently supported beyond the controlled setting.
The evaluation is largely in-distribution and conducted on a narrow benchmark. While the results suggest that the approach works in this specific setting, they do not provide clear evidence for broader generalization or research-level insight, particularly in relation to more recent multimodal models or stronger adaptation baselines.